# Amyloid-Beta Co-Pathology Is a Major Determinant of the Elevated Plasma GFAP Values in Amyotrophic Lateral Sclerosis

**DOI:** 10.3390/ijms241813976

**Published:** 2023-09-12

**Authors:** Andrea Mastrangelo, Veria Vacchiano, Corrado Zenesini, Edoardo Ruggeri, Simone Baiardi, Arianna Cherici, Patrizia Avoni, Barbara Polischi, Francesca Santoro, Sabina Capellari, Rocco Liguori, Piero Parchi

**Affiliations:** 1Dipartimento di Scienze Biomediche e Neuromotorie, Alma Mater Studiorum Università di Bologna, 40139 Bologna, Italy; andrea.mastrangelo4@studio.unibo.it (A.M.); simone.baiardi6@unibo.it (S.B.); patrizia.avoni@unibo.it (P.A.); sabina.capellari@unibo.it (S.C.); rocco.liguori@unibo.it (R.L.); 2IRCCS Istituto delle Scienze Neurologiche di Bologna, 40139 Bologna, Italy; veriavacchiano@gmail.com (V.V.); corrado.zenesini@isnb.it (C.Z.); edoardo.ruggeri@ausl.bologna.it (E.R.); arianna.cherici@ausl.bologna.it (A.C.); barbara.polischi@isnb.it (B.P.); francesca.santoro@isnb.it (F.S.)

**Keywords:** amyotrophic lateral sclerosis, Alzheimer’s disease, GFAP, biofluid biomarkers, co-pathology, neurodegeneration, neuropsychology

## Abstract

Recent studies reported increased plasma glial acidic fibrillary protein (GFAP) levels in amyotrophic lateral sclerosis (ALS) patients compared to controls. We expanded these findings in a larger cohort, including 156 ALS patients and 48 controls, and investigated the associations of plasma GFAP with clinical variables and other biofluid biomarkers. Plasma GFAP and Alzheimer’s disease (AD) cerebrospinal fluid (CSF) biomarkers were assessed by the single molecule array and the Lumipulse platforms, respectively. In ALS patients, plasma GFAP was higher than in controls (*p* < 0.001) and associated with measures of cognitive decline. Twenty ALS patients (12.8%) showed a positive amyloid status (A+), of which nine also exhibited tau pathology (A+T+, namely ALS-AD). ALS-AD patients showed higher plasma GFAP than A− ALS participants (*p* < 0.001) and controls (*p* < 0.001), whereas the comparison between A− ALS and controls missed statistical significance (*p* = 0.07). Plasma GFAP distinguished ALS-AD subjects more accurately (area under the curve (AUC) 0.932 ± 0.027) than plasma p-tau181 (AUC 0.692 ± 0.058, *p* < 0.0001) and plasma neurofilament light chain protein (AUC, 0.548 ± 0.088, *p* < 0.0001). Cognitive measures differed between ALS-AD and other ALS patients. AD co-pathology deeply affects plasma GFAP values in ALS patients. Plasma GFAP is an accurate biomarker for identifying AD co-pathology in ALS, which can influence the cognitive phenotype.

## 1. Introduction

Amyotrophic lateral sclerosis (ALS) is a neurodegenerative disorder primarily characterized by the loss of motor neurons in the brain and the spinal cord, thus leading to progressive muscle weakness and wasting and eventually to death.

However, the clinical spectrum of ALS is not limited to motor abnormalities, with up to half of patients displaying cognitive and/or behavioral impairment at different stages of severity and around 10–15% of subjects fulfilling the criteria for full-blown frontotemporal dementia (FTD). Moreover, the severity of cognitive decline seems to worsen with the progression of the disease, similar to motor impairment, further contributing to the disability of patients [1].

ALS is mainly a sporadic disease, but familiar forms account for 5–10% of cases. More than 20 disease-causing genes have been described, with the repeat expansion of the *C9Orf72* gene being the most common genetic abnormality reported.

Neuropathologically, ALS is a TDP-43 proteinopathy characterized by TDP-43-enriched inclusions in affected neurons. However, due to the high disease prevalence, a significant proportion of ALS patients develop secondary Alzheimer’s disease (AD) pathological changes of various severities, which may also contribute to cognitive impairment in these patients [2,3].

Biofluid biomarkers are urgently needed in the ALS field to improve the diagnostic accuracy in vitam, predict and track the disease progression, and monitor the response to potential disease-modifying agents. In this regard, the neurofilament light chain protein (NfL), a biomarker of unspecific neuroaxonal damage, has recently shown excellent diagnostic and prognostic value in both blood [4,5] and cerebrospinal fluid (CSF) [5,6] and appears to be a promising marker of ongoing neurodegeneration in clinical-pharmacological trials [7]. Glial activation and neuroinflammation markers are also increasingly exploited in neurodegenerative diseases, given their relevance to the pathogenesis of many neurological disorders [8,9]. Since there is an increased expression of the glial fibrillary acidic protein (GFAP), the main component of the intermediate filaments, in activated astrocytes, and its spillover in the extracellular space increases following astrocyte damage [10,11], GFAP levels have been recently explored in both CSF and blood of patients with different central nervous system (CNS) disorders. Most significantly, plasma GFAP levels have been shown to be considerably higher in patients with AD than in other diseases associated with dementia, even in a prodromal or asymptomatic phase [12,13,14,15]. 

As for ALS, data on CSF and blood GFAP levels are fewer and less concordant, with a preliminary study showing elevated CSF GFAP values in ALS patients compared to controls [16] and others reporting no difference in blood or CSF between ALS and healthy subjects [17,18]. Recently, a single study showed higher blood GFAP values in ALS than in controls and reported a positive correlation between biomarker levels and parameters of cognitive and behavioral impairment [19].

In the present study, we compared plasma GFAP levels in the most extensive ALS cohort examined to date with those of patients with frontotemporal dementia (FTD) and neurological controls. Furthermore, we evaluated the association of plasma GFAP values with clinical variables and plasmatic and CSF levels of other biofluid biomarkers, including those reflecting AD pathology. Finally, we studied the value of plasma GFAP in predicting survival in ALS patients.

## 2. Results

### 2.1. Distribution of Plasma GFAP Level Values across the Diagnostic Categories and Clinical Correlates of Plasma GFAP in ALS Patients

The demographic variables of ALS patients and controls and clinical features of the ALS cohort are detailed in Table 1 and Table 2.

Age at sampling (*p* = 0.07) and sex distribution (*p* = 0.73) were not significantly different between ALS patients and controls.

ALS patients showed higher plasma GFAP levels than controls (*p* = 0.0004) (Figure 1a).

Plasma GFAP levels were not significantly different across onset types (*p* = 0.52), clinical phenotypes (*p* = 0.65), King’s stages (*p* = 0.52), genetic status (*p* = 0.59), and the number of regions with upper motor neuron (UMN) (*p* = 0.07) or lower motor neuron (LMN) signs (*p* = 0.57) or both (*p* = 0.07). A slight increase in plasma GFAP levels in ALS females compared to males almost reached statistical significance (179.43 (126.4–238) vs. 152.19 (110.4–231.9), *p* = 0.052).

Using regression analysis, we found that GFAP levels were significantly influenced by age at both onset (β = 0.026, 95% CI 0.02 to 0.032, *p* < 0.0001) and sampling (β = 0.026, 95% CI 0.020 to 0.032, *p* < 0.001), Revised Amyotrophic Lateral Sclerosis Functional rating (ALSFRS-R) scale (β = −0.031, 95% CI −0.046 to −0.017, *p* < 0.001), disease progression rate (DPR) (β = 0.119, 95% CI 0.023 to 0.217, *p* = 0.016), forced vital capacity (FVC) values (β = −0.004, 95% CI −0.008 to −0.001, *p* = 0.02), and creatinine levels (β = 0.699, 95% CI 0.310 to 1.088, *p* = 0.0005).

In contrast, GFAP values were not related to disease duration (*p* = 0.7), Medical Research Council (MRC) score (*p* = 0.36), creatine phosphokinase (CPK) levels (*p* = 0.095), albumin index (*p* = 0.2), and body mass index (BMI) (*p* = 0.28). A multivariable linear regression model after adjusting for covariables confirmed that GFAP values were significantly independently influenced by age at sampling (β = 0.019, 95% CI 0.013 to 0.026, *p* < 0.001), creatinine (β = 0.573, 95% CI 0.239 to 0.906, *p* = 0.001), ALSFRS-R scale (β = −0.023, 95% CI −0.039 to −0.007, *p* = 0.004), and A status (β = 0.333, 95% CI 0.101 to 0.565, *p* = 0.005).

### 2.2. Association of Plasma GFAP with Measures of Cognitive Impairment in ALS Patients

Plasma GFAP did not statistically differ among ALS patients belonging to different Strong’s Categories (*p* = 0.16) but was higher in ALS patients with associated FTD (230.7 (154–317.9) vs. 157.8 (116.6–225.25), *p* = 0.042) as compared to pure motor ALS.

Plasma GFAP levels significantly differed among ALS-FTD, pure FTD, and pure motor ALS patients (namely without clinical signs of FTD) with a negative A status (*p* = 0.001), with the post-hoc analysis revealing significantly higher levels in pure FTD (199.0 (132.3–293.9)) than in pure motor ALS subjects (n = 125) (152.2 (111.3–197.3), *p* = 0.001).

The GFAP levels correlated with ALS-specific subscores of Edinburgh Cognitive and Behavioral ALS Screen (ECAS) (Rho = −0.22, *p* = 0.04), Brief Mental Deterioration Battery (BMDB) total score (Rho = −0.23, *p* = 0.019), Category Fluency scores (Rho = −0.20, *p* = 0.036), and Freehand copy of drawings (Rho = −0.26, *p* = 0.01). A trend of significance was observed with the ECAS total score (Rho = −0.20, *p* = 0.06), ECAS executive functions (Rho = −0.19, *p* = 0.07), ECAS memory (Rho = −0.2, *p* = 0.06), Letter Fluency scores (Rho = −0.17, *p* = 0.07). No correlations were found with other ECAS subscores and other neuropsychological tests. 

The association of plasma GFAP with BMDB total score (Rho = −0.20, *p* = 0.048) and Freehand copy of drawings (Rho = −0.24, *p* = 0.02) was retained after excluding ALS patients with a positive CSF amyloid profile.

### 2.3. Association of Plasma GFAP with Other Plasma and CSF Biomarkers in ALS Patients

In ALS patients, a moderate inverse correlation was found between plasma GFAP and CSF Aβ ratio (Rho = −0.34, *p* < 0.001), which was consistent even after accounting for age at sampling (β = −0.84; *p* < 0.001). Plasma GFAP was more weakly associated with plasma NfL (Rho = 0.30, *p* = 0.0001), CSF t-tau (Rho = 0.27, *p* = 0.004), plasma p-tau181 (Rho = 0.25, *p* = 0.001), and CSF p-tau (Rho = 0.23, *p* = 0.004). There was no association between plasma GFAP and CSF NfL (Rho = 0.09, *p* = 0.25), even after accounting for plasma creatinine (*p* = 0.23) or disease duration (*p* = 0.11).

### 2.4. Plasma GFAP Levels and Clinical Variables According to A and T Status in ALS Patients

Due to the moderate association between plasma GFAP and CSF Aβ ratio, we stratified ALS patients according to their A and T status.

Twenty ALS patients (12.8%) showed a positive amyloid status (A+), and nine of them (5.8% of the whole ALS cohort) had a CSF profile also suggestive of p-tau deposition (A+T+ profile). At sampling, A+ ALS patients were significantly older than those A− (74.5 (70.2–81.5) vs. 64.0 (55.0–71.0), *p* < 0.0001).

Plasma GFAP significantly differed among A+T+, A+T−, A−ALS patients and controls (*p* < 0.0001), with each A+ ALS subgroup showing higher values than controls (A+T+ vs. controls, *p* < 0.0001; A+T− vs. controls, *p* = 0.0003) and A− subjects (A+T+ vs. A−, *p* < 0.0001; A+T− vs. A−, *p* = 0.02). Plasma GFAP did not significantly differ between A+T+ and A+T− ALS patients (*p* > 0.99), while the comparison between A−ALS patients and controls reached a trend of significance (*p* = 0.07) (Figure 1b).

Biomarker values in ALS patients stratified by their A and T status are reported in Table 3.

Accordingly, plasma GFAP yielded a high value in the discrimination between A+ and A−ALS patients (AUC 0.847 ± 0.041), significantly higher than that of other plasma biomarkers (plasma p-tau181 AUC 0.706 ± 0.048, plasma GFAP vs. plasma p-tau181, *p* = 0.008; plasma NfL AUC 0.528 ± 0.064, plasma GFAP vs. plasma NfL, *p* = 0.0003) and comparable to that of CSF p-tau (CSF p-tau AUC 0.875 ± 0.038, plasma GFAP vs. CSF p-tau *p* = 0.52) (Table 4, Figure 2a–c).

Plasma GFAP showed very high accuracy (AUC 0.932 ± 0.027) in discriminating A+T+ patients (AD/ALS) from those not displaying a CSF profile consistent with a full-blown AD pathology (not-AD/ALS, namely A− and A+T− ALS patients), which was significantly higher than that of any other plasma biomarker (plasma p-tau181 AUC 0.692 ± 0.058, plasma GFAP vs. plasma p-tau181 *p* = 0.0008; plasma NfL AUC 0.548 ± 0.088, plasma GFAP vs. plasma NfL *p* < 0.0001) (Table 4, Figure 2d–f).

In comparison to not-AD/ALS, AD/ALS patients showed significantly lower scores at the ECAS battery (ECAS total equivalent scores, *p* = 0.04), at the Mini-Mental State Examination (MMSE) test (albeit not age- and education-adjusted) (*p* = 0.03), and at neuropsychological tests exploring short-term visual memory (*p* = 0.01) (Table 5). A trend of significance was also found for ECAS ALS-specific equivalent scores (*p* = 0.06).

The full list of scores obtained by ALS patients at the different neuropsychological tests is reported in Appendix A.

### 2.5. Prognostic Value of Plasma GFAP in ALS Patients

Univariable Cox regression analysis (156 patients with ALS, 77 dead) identified as prognostic factors the following clinical variables: age at onset (*p* = 0.005); ALSFRS-R scale (*p* < 0.001); DPR (*p* < 0.001); bulbar onset (*p* = 0.001); FTD status (*p* = 0.048). As for biomarkers, plasma GFAP (Hazard Ratio (HR) 2.46, *p* < 0.001), plasma NfL (HR 1.01, *p* < 0.001), and plasma p-tau181 (HR 1.11, *p* = 0.02) were identified as predictors of survival (Appendix A, Figure 3).

However, the independent prognostic value of plasma GFAP (*p* = 0.64) was not confirmed in the Multivariate Cox regression analysis including clinical variables (Appendix A). In the multivariable analysis including plasma biomarkers, plasma GFAP (*p* = 0.032) and both plasma NfL (*p* < 0.001) and p-tau181 (*p* = 0.042) independently predicted survival in ALS patients (Table 6).

## 3. Discussion

In this work, we investigated the distribution of plasma GFAP levels in an extensive cohort of deeply phenotyped ALS patients and explored their clinical and neuropsychological correlates.

In ALS patients, plasma GFAP values were significantly higher than in controls, correlated with age at sampling, in line with previous reports [18,19], and showed slightly increased values in females than in males.

Plasma GFAP was moderately associated with the ALSFRS-R scale and DPR but did not show any relationship with other parameters of disease severity or extent of motor impairment, such as King’s stage, MRC score, or the number of regions displaying UMN or LMN signs. Similarly, we found no differences in plasma levels across onset types and clinical phenotypes. Taken together, these data, in agreement with previous reports in smaller ALS cohorts [18,19], suggest that plasma GFAP elevation in ALS also reflects the astrocytic activation secondary to neurodegeneration at sites unrelated to motor neurons. The moderate association between plasma GFAP and parameters of cognitive impairment, including ALS-specific ECAS scores, the BMDB total score, and the scores in tests exploring semantic fluency and constructional praxis, suggest a link with extra-motor cortical areas. Accordingly, plasma GFAP was significantly elevated in ALS patients displaying a full-blown FTD despite the lack of a significant difference in the biomarker levels across the Strong classification categories, probably due to the subgroups’ scarce numerosity. Furthermore, plasma GFAP levels were significantly higher in pure-FTD patients than those with pure motor ALS. 

Increasing evidence suggests an extra-motor involvement in ALS [1], including neuropathological studies [20] showing astrocyte activation or degeneration in brain areas different from those harboring motor neurons. Notably, preliminary data [21], albeit not confirmed by other authors [2], indicate a higher representation of reactive astrocytes in brain areas relevant for superior functions in ALS patients with cognitive impairment than in pure motor ALS.

In this scenario, the unique association of the biomarker with the ALSFRS-R scale and DPR could reflect the correlation with the spreading process of the disease, possibly driven by the correlation with age, with older patients typically showing a more severe disease.

The results of our survival analysis in ALS patients align with these observations. Indeed, plasma GFAP significantly predicted survival in the univariate analysis. Still, the significance was lost when covarying with well-known prognostic clinical factors in ALS, such as type of onset and the ALSFRS-R scale. These data reflect the lack of correlation of plasma GFAP with scores of motor impairment severity, which plays the most important role in determining the disease course. Notably, plasma GFAP retained its prognostic value when we only accounted for plasma NfL, and p-tau181, which were previously shown to predict survival in ALS patients [4,5,22]. This confirms the specific prognostic contribution of this biomarker, possibly indicating cognitive impairment, and suggests that prognostic estimates based on different blood biomarkers, with p-tau181 mainly reflecting LMN degeneration and NfL expressing the overall disease severity, may have an added value in ALS patients.

As an important finding, we showed for the first time that plasma GFAP levels in ALS patients are significantly influenced by AD co-pathology. In detail, plasma GFAP was moderately associated with the CSF Aβ ratio, even after correction for age. Moreover, when stratifying patients according to amyloid status, A+ subjects, independently from their T status, showed significantly higher GFAP values than A−ALS patients and controls. Interestingly, in the multiple-group comparison, the biomarker’s values were not significantly different between controls and A−ALS, probably reflecting the relatively low degree of astrogliosis found in ALS patients’ brains compared to that of subjects with AD.

Plasma GFAP levels, more accurately than those in CSF, have been shown to distinguish patients with underlying amyloid pathology independently from the severity of cognitive impairment and even in patients with a primary alternative neurodegenerative disorder, such as Lewy Body disease [12,13,14,23]. This probably reflects the strong relationship between activated astrocytes and amyloid plaques in AD patients’ brains [15,24]. In this view, plasma GFAP could serve as a valid surrogate blood biomarker for the identification of AD co-pathology in ALS patients, given the suboptimal value of plasma p-tau181 in these subjects due to its likely peripheral source, as already demonstrated in two studies [22,25] and confirmed in this work in a larger cohort. Plasma GFAP showed the highest accuracy among the examined plasma biomarkers in identifying ALS patients with positive amyloid status and full-blown AD pathology. Interestingly, plasma GFAP values were not significantly different between ALS A+T− and A+T+, further indicating that astrogliosis, so GFAP elevation in blood, is an initial event in the AD pathogenetic cascade, as already supported by biomarkers studies in autosomal dominant AD mutation carriers [26].

Extensive studies on the prevalence of AD co-pathology in ALS patients are lacking, with some authors reporting a 20% prevalence, likely age-related [2], while others show a higher percentage of AD neuropathological changes, mainly in subjects with cognitive decline [3]. In our cohort, although only through a biofluid-biomarker-based approach, we reported an amyloid co-pathology in approximately 13% of ALS patients (only 6% with both A and T positive status), which is in line with the estimates of amyloid deposition prevalence in the age-matched general population [27] and therefore not supporting a causal connection between AD and ALS pathologies.

Albeit only preliminary, our data seem to support a different cognitive profile in ALS patients with concomitant AD co-pathology, with these latter showing significantly lower scores in multi-domain scales (ECAS total equivalent scores, MMSE) and in specific cognitive domains, such as visual memory, typically impaired at early stages in the AD continuum [28]. Similarly, the association of plasma GFAP with scores of semantic fluency and constructional praxis may be interpreted in this view.

In summary, plasma GFAP, being strictly associated with amyloid co-pathology in ALS, could serve as a biomarker of cognitive impairment in ALS and aid in identifying patients with cognitive features atypical for ALS-FTD dementia. Further studies are needed to show more detailed differences in the cognitive profile of AD/ALS subjects.

Regarding the possible influence of gender on plasma GFAP values, we found that in our ALS population females showed higher biomarker levels than males, with the difference almost reaching statistical significance. Higher plasma GFAP values in females were previously reported in ALS patients, albeit potentially related to the older age [19], and in subjects with other neurodegenerative disorders as well [12,13]. However, such a difference was not highlighted in other patient groups [22]. In our cohort, the slightly higher plasma GFAP values in females could be at least partially related to the higher prevalence of beta-amyloid co-pathology (A+, females 15.5%, males 11.2%). Given the overall inconclusive data, further studies are required to fully explore the influence of gender on the distribution of plasma GFAP values.

The moderate association of GFAP levels with plasma creatinine deserves further comments. The relationship between renal function and levels of plasma biomarkers, including GFAP, has already been reported [19,29]. However, in one of these studies, a significant overall effect of creatinine on the accuracy of using plasma biomarker levels to predict the risk of conversion to dementia in AD patients could not be demonstrated [29]. Nonetheless, given the high prevalence of chronic kidney disease in the general population, especially in the elderly, further studies are required to clarify the influence of renal function on plasma GFAP levels and their clinicopathological correlates.

The inclusion of a large sample of deeply characterized ALS patients and a high number of different CSF and plasma biomarkers available is the main strength of our work. Secondarily, the deep categorization of patients’ cognitive impairment through an extensive battery of neuropsychological tests, including the specific battery validated for ALS, (i.e., ECAS), is another added value. On the contrary, the lack of a systematized evaluation of the impact of comorbidities and medication on GFAP values in ALS patients is one of the limitations, as the lack of neuropathological correlates of GFAP elevation in our cohort and the relatively low number of ALS patients with a full-blown AD co-pathology, partially due to the rarity of ALS itself. Further studies involving neuropathological cohorts are required to confirm our results and address the relationship between plasma GFAP levels and the burden of AD co-pathology in ALS patients.

## 4. Materials and Methods

### 4.1. Inclusion Criteria and Clinical Assessment

We included 156 patients diagnosed with ALS according to the Revised El Escorial criteria [30], evaluated at the Institute of Neurological Sciences of Bologna between September 2014 and December 2022. All patients had baseline CSF and plasma samples available. We also separately studied 50 patients with a clinical diagnosis of FTD according to international criteria [31,32], without any signs of UMN or LMN impairment (pure FTD) and a negative CSF amyloid profile. Finally, 48 subjects without clinical evidence of neurological disease were also included as controls.

For ALS patients, we collected the following clinical variables at baseline: age at onset; sex; disease duration (time elapsed between the disease onset and CSF/plasma sampling); type of onset; ALSFRS-R; MRC scale of 0 to 5 [5]; FVC; BMI; and King’s clinical stage. Patients were subdivided according to a validated classification [33] into the following phenotype categories: classic, bulbar, respiratory, UMN-predominant (PUMN), primary lateral sclerosis (PLS), flail arm syndrome, flail leg syndrome, and progressive muscular atrophy (PMA). However, to reach sufficient statistical power for comparisons, we also grouped patients into main categories, i.e., classic (including respiratory), bulbar, PUMN (i.e., PUMN and PLS), and LMN-predominant (PLMN, including flail arm/leg and PMA).

One hundred and fifty-three ALS patients performed genetic analysis, including the screening for mutations in the most frequent ALS-related genes (i.e., *SOD1*, *FUS*, *TARDBP*, and the *C9Orf72* repeats expansion) [34]. Furthermore, the apolipoprotein E (*APOE*) genotype was analyzed, and *APOE* ε4 carriers were defined as individuals with at least one *APOE* ε4 allele.

Cognitive status was evaluated through a neuropsychological assessment encompassing executive function, memory, visuospatial function, language, and social cognition domains. The battery included the MMSE, the Frontal assessment battery (FAB) [35], the Letter Fluency Test (FAS); the Category Fluency Test; the BMDB [36], and the ECAS [37]. For this latter, we computed the five cognitive domains of executive functions, verbal fluency, language, memory, and visuospatial functions, composite ALS-specific (i.e., executive + verbal fluency + language) and ALS-nonspecific (i.e., memory + visuospatial) subscores. ECAS scores were adjusted for age and education, as previously reported [38].

Patients were classified accordingly into five categories (purely motor ALS (ALS-CN), ALS with cognitive impairment (ALSci), ALS with behavioral impairment (ALSbi), and ALS with cognitive and behavioral impairment (ALScbi), FTD) [39]. To enable statistical analysis with sufficient power, we grouped ALSbi and ALSci categories. We also used a binary classification (ALS-FTD or pure motor ALS patients), according to the presence of FTD only, as clinically assessed [31].

UMN involvement was scored by the number of regions (bulbar, cervical, and lumbosacral region) showing UMN signs at clinical assessment. In contrast, we used clinical and electromyography (EMG) assessments to establish the extent of LMN involvement according to the Awaji criteria [40]. The DPR was calculated using the following formula: (48−ALSFRS-R score at the time of sampling)/months elapsed between disease onset and sampling) and patients were accordingly divided into slow (DPR < 0.5), intermediate (DPR 0–5–1), and fast progressors (DPR > 1) [5]. Patients performed routine laboratory blood examinations, among which we collected serum creatinine, CPK, and serum albumin.

None of the ALS patients were under Riluzole treatment at the time of sampling.

### 4.2. CSF and Plasma Analyses

EDTA plasma samples were collected, aliquoted, and stored at −80 °C, according to standard procedures. CSF samples were obtained by lumbar puncture following a routine procedure, centrifuged in case of blood contamination (even minimal), divided into aliquots, and stored in polypropylene tubes at −80 °C until analysis.

From CSF routine analysis, we extrapolated CSF albumin to calculate the albumin index.

Plasma GFAP, Plasma NfL, and CSF NfL levels were determined with the Single molecule array (Simoa) technology [41] on a Simoa SR-X instrument using the commercially available GFAP Discovery and NF-light Advantage Kits (Quanterix, Billerica, MA, USA). The mean intra- and inter-assay coefficients of variation (CVs) were below 15% for all analyses.

CSF Aβ42, Aβ40, p-tau, and t-tau were measured by automated chemiluminescent enzyme immunoassay on the Lumipulse G600II platform (Fujirebio, Gent, Belgium). The inter-assay CVs were <8% for all biomarkers. The Aβ42/Aβ40 was calculated as described [42]. We used in-house validated cutoffs to determine pathological values for the AD core markers. In particular, a CSF Aβ42/Aβ40 ratio < 0.68 was considered supportive of amyloid deposition (i.e., A+ according to the ATN classification [43]), while a CSF phosphorylated tau at site 181 (p-tau181) > 62 pg/mL was considered indicative of p-tau deposition (i.e., T+).

### 4.3. Statistical Analysis 

Statistical analyses were performed using Stata SE V.14.2 (StataCorp, College Station, TX, USA) and GraphPad Prism V.7 (GraphPad Software, La Jolla, CA, USA) software. For continuous variables, the Mann–Whitney or the Kruskal–Wallis test (followed by the Dunn–Bonferroni post hoc test) was used for comparisons between groups. Fisher’s test was applied for categorical variables. We used Spearman’s Rho coefficient to test the correlation between plasma GFAP levels and other CSF/plasma biomarkers (i.e., plasma p-Tau181, CSF/plasma NfL) and age- and education-adjusted scores from neuropsychological tests.

The association between plasma GFAP levels and clinical variables was assessed using univariable and multivariable models with the log-transformed plasma GFAP values as dependent variables and the clinical variables as independent variables. In the multivariable models, we adjusted for age at sampling, ALSFRS-R scale, A status, DPR, FVC, and creatinine. The results are presented as ß coefficients and 95% Confidence Interval (95% CI).

Receiver operating characteristic (ROC) analyses were performed to establish the accuracy of different plasma biomarkers in the discrimination of ALS patients according to their A and T status. ROC curves were compared through the DeLong test. The optimal cut-off value for each biomarker was defined using the maximized Youden Index.

For the prognostic analysis, the Kaplan–Meier estimate calculated the cumulative time-dependent probability of death. The time of entry into the analysis was the date of the first sampling, and the endpoint was the date of death/tracheostomy or the date of the last follow-up information, whichever came first. Univariable and multivariable Cox regression models were performed to study prognostic factors in ALS. In detail, we performed two separate multivariable analyses: one including plasma GFAP and clinical variables (age at onset, type of onset, ALSFRS-R score, presence of FTD, DPR) and the other one with other plasma biomarkers with known prognostic value (i.e., plasma GFAP, NfL, and p-tau181). The results are presented as Hazard Ratios (HR) and 95% CI. The assumption of proportional hazard was assessed by Schoenfeld residuals. Differences were considered significant at *p* < 0.05.

## 5. Conclusions

In conclusion, our work provides evidence that plasma GFAP is elevated in ALS patients compared to controls, but this elevation is mainly affected by concomitant amyloid-beta pathology. Plasma GFAP shows the highest accuracy among the most common plasma biomarkers in identifying AD co-pathology in ALS and is related to measures of cognitive impairment in ALS patients. Finally, including plasma GFAP in survival multivariable analyses with other plasma biomarkers could add value to the prognosis estimation of ALS patients.

## Figures and Tables

**Figure 1 ijms-24-13976-f001:**
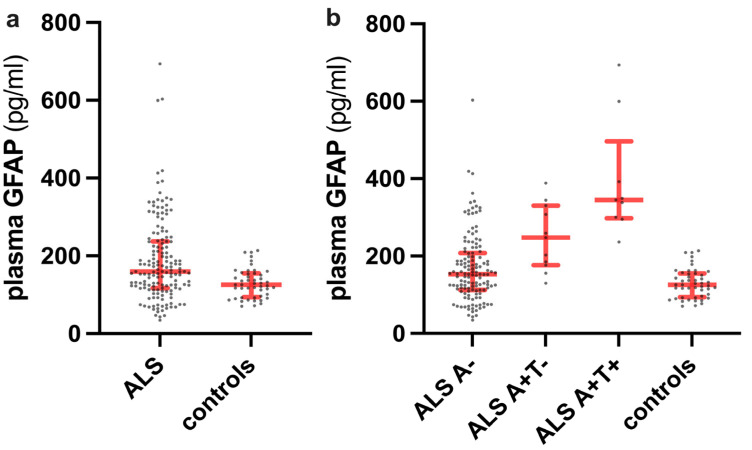
Plasma GFAP levels in the whole ALS cohort compared to controls (**a**) and in the ALS patients stratified by A and T status (**b**).

**Figure 2 ijms-24-13976-f002:**
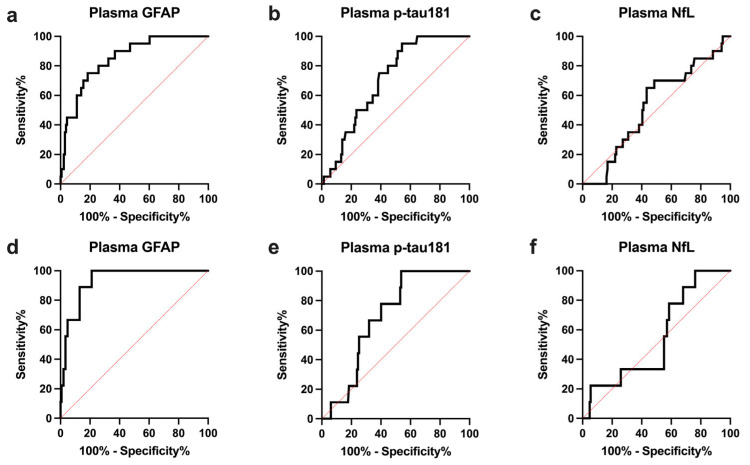
ROC curves of plasma biomarkers in the discrimination of ALS patients with amyloid co-pathology (A+ status) (**a**–**c**) and ALS patients with concomitant full-blown AD pathology (A+T+ status) (**d**–**f**). GFAP, glial fibrillary acidic protein; NfL, neurofilament light chain; p-tau181, plasma phosphorylated tau protein 181; ROC, receiver operating characteristic.

**Figure 3 ijms-24-13976-f003:**
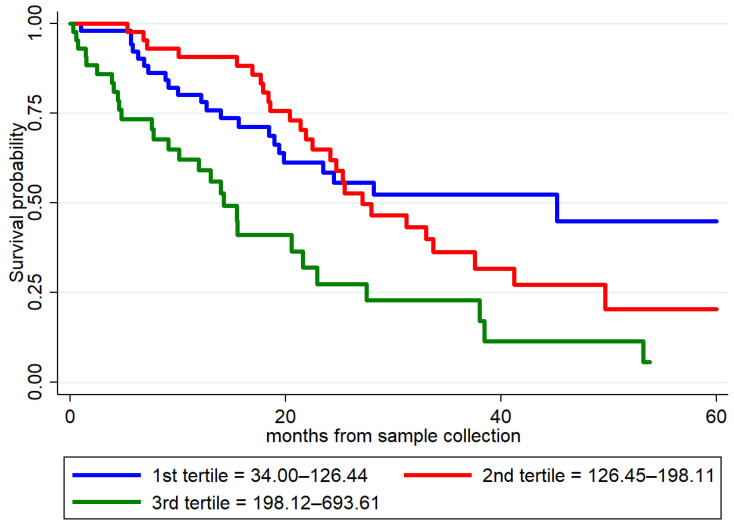
Survival curves in patients with ALS according to the values of plasma GFAP. Biomarker levels were stratified into low, mid, and high tertiles and are expressed in pg/mL. GFAP, glial fibrillary acidic protein.

**Table 1 ijms-24-13976-t001:** Demographic variables and biomarkers values in ALS patients and controls.

	ALS Patients (n = 156)	Controls (n = 48)	*p* Values
Age at sampling, years *	66.0 (56.0–72.0)	61.0 (60.0–64.0)	0.07 ^†^
Female, n (%)	58 (37.2)	16 (33.3)	0.73 ^§^
Plasma GFAP, pg/mL *	159.70 (117.30–236.70)	125.9 (93.52–154.70)	**0.0004 ^†^**
Plasma NfL, pg/mL *	70.50 (41.25–113.70)	10.76 (9.41–15.71)	**<0.0001 ^†^**
Plasma p-tau181, pg/mL *	2.71 (1.74–4.96)	0.99 (0.76–1.36)	**<0.0001 ^†^**

*: Data are expressed as median (interquartile range). ^†^ Kruskal–Wallis test; ^§^ Fisher’s test. Significant *p*-values are reported in bold. Abbreviations: ALS, amyotrophic lateral sclerosis; GFAP, glial fibrillary acidic protein; NfL, neurofilament light chain; p-tau181, plasma phosphorylated tau 181.

**Table 2 ijms-24-13976-t002:** Clinical features in ALS patients.

ALS Patients (n = 156)	
	N (%)
**Type of onset**	
Bulbar	34 (21.8)
Spinal	105 (67.3)
Pseudopolyneuritic	12 (7.7)
Pyramidal	5 (3.2)
**Clinical phenotype**	
Classic	92 (58.9)
Bulbar	22 (14.1)
Respiratory	1 (0.6)
PUMN	11 (7.0)
PLS	3 (1.9)
Flail arm syndrome	10 (6.4)
Flail leg syndrome	7 (4.5)
PMA	10 (6.4)
**Diagnostic categories**	
Definite ALS	27 (17.3)
Probable ALS	58 (37.2)
Probable laboratory-supported ALS	35 (22.4)
Possible ALS	26 (16.7)
Unclassified (PMA)	10 (6.4)
**King’s staging**	
1	8 (5.1)
2	53 (34.0)
3	83 (53.2)
4	12 (7.7)
**Strong’s categories (n = 128)**	
ALS-CN	77 (60.1)
ALSbi	21 (16.4)
ALSci	7 (5.5)
ALScbi	9 (7.0)
ALS-FTD	14 (10.9)
**Genetic status (n = 153)**	
*C9Orf72* RE carriers	15 (9.8)
*SOD1* mutation carriers	3 (2.0)
*TARDBP* mutation carriers	1 (0.6)
Wild-type	134 (87.6)
Deceased/with tracheostomy	77 (49.3)
	**Median (IQR)**
Disease duration (months)	13 (8–24)
ALSFRS-R scale (n = 154)	41 (38.0–44.0)
MRC score (n = 155)	4.6 (4.25–4.8)
FVC * (n = 140)	90.0 (74.5–106.0)
BMI (n = 146)	24.6 (22.1–27.7)
Creatinine	0.74 (0.65–0.85)
CPK (n = 155)	197 (120–379)
Blood-brain barrier index (n = 153)	7.0 (5.5–10.6)

* Expressed as a percentage of the predicted volume. If not otherwise specified, data are available for the whole ALS cohort. Abbreviations: ALS, amyotrophic lateral sclerosis; ALSbi, amyotrophic lateral sclerosis with behavioral impairment; ALScbi, amyotrophic lateral sclerosis with combined cognitive and behavioral impairment; ALSci, amyotrophic lateral sclerosis with cognitive impairment; ALS-CN, cognitively normal amyotrophic lateral sclerosis; ALSFRS-R, Revised Amyotrophic Lateral Sclerosis Functional rating scale; BMI, body mass index; CPK, creatine phosphokinase; FVC, forced vital capacity; FTD, frontotemporal dementia; IQR, interquartile range; MRC, Medical Research Council; PLS, primary lateral sclerosis; PMA, progressive muscular atrophy; PUMN, prevalent upper motor neuron; RE, repeats expansion.

**Table 3 ijms-24-13976-t003:** Biomarkers values and *APOE* status in ALS patients stratified by their A and T status.

	ALS, A+T+ (n = 9)	ALS, A+T− (n = 11)	ALS, A− (n = 136)	*p* Values
Plasma GFAP, pg/mL *	345.0 (297.4–496.2)	247.4 (176.2–330.0)	153.0 (112.2–207.6)	**<0.0001 ^†^**
Plasma NfL, pg/mL *	66.21 (52.95–172.10)	55.20 (31.30–103.0)	73.50 (41.25–115.0)	0.57 ^†^
Plasma p-tau181, pg/mL *	4.72 (2.87–5.66)	3.79 (2.81–6.46)	2.55 (1.56–4.63)	**0.01 ^†^**
CSF p-tau181, pg/mL *	82.00 (73.25–96.55)	45.90 (41.30–52.60)	32.65 (26.70–42.13)	**<0.0001 ^†^**
CSF t-tau, pg/mL *	585.0 (455.5–635.0)	286.0 (265.0–358.0)	255.5 (204.8–357.0)	**<0.0001 ^†^**
CSF NfL, pg/mL *	6135 (3737–10,570)	2618 (1731–6390)	6102 (3165–10,844)	0.21 ^†^
*APOE* ε4 carriers, positive (%)	3 (33.3)	4 (36.4)	15 (11.0)	**0.01 ^§^**

* Expressed as median (interquartile range); ^†^ Kruskal–Wallis test; ^§^ Fisher’s test. Significant *p*-values are reported in bold. Abbreviations: *APOE*, apolipoprotein E; ALS, amyotrophic lateral sclerosis; CSF, cerebrospinal fluid; GFAP, glial acidic fibrillary protein; NfL, neurofilament light chain; p-tau181, plasma phosphorylated tau 181; t-tau, total tau protein.

**Table 4 ijms-24-13976-t004:** Value of different plasma and CSF biomarkers in the discrimination of ALS patients according to their A and T status.

ALS A+ vs. ALS A−					
	AUC (95% CI)	*p*-Value *	Sensitivity, % (95% CI)	Specificity, % (95% CI)	Optimal Cutoff Value ^†^
Plasma GFAP	0.847 (0.766–0.929)	-	75.0 (53.1–88.8)	81.6 (74.3–87.2)	>236.3
Plasma p-tau181	0.706 (0.611–0.800)	**0.0008**	95.0 (76.4–99.7)	45.6 (37.4–54.0)	>2.22
Plasma NfL	0.528 (0.403–0.653)	**0.0003**	65.0 (43.3–81.9)	56.6 (48.2–64.5)	<67.0
CSF p-tau181	0.875 (0.801–0.949)	0.52	90.0 (69.9–98.2)	75 (67.1–81.5)	>41.15
**AD/ALS vs. not-AD/ALS**					
	**AUC** **(95% CI)**	** *p* ** **-Value ***	**Sensitivity, %** **(95% CI)**	**Specificity, %** **(95% CI)**	**Optimal Cutoff Value ^†^**
Plasma GFAP	0.932 (0.879–0.985)	-	100 (70.1–100)	78.9 (71.6–84.7)	>236.3
Plasma p-tau181	0.692 (0.578–0.807)	**0.0008**	100 (70.1–100)	46.2 (38.4–54.3)	>2.47
Plasma NfL	0.548 (0.376–0.721)	**<0.0001**	100 (70.1–100)	23.8 (17.6–31.3)	>38.1

* Comparison with the AUC of plasma GFAP (DeLong Test); ^†^: expressed in pg/mL and calculated through the Youden Index. AD/ALS patients showed an A+T+ CSF profile. Not-AD/ALS subjects did not show a CSF profile consistent with a full-blown AD pathology (namely A− and A+T−). Significant *p*-values are reported in bold. Abbreviations: AD, Alzheimer’s disease; ALS, amyotrophic lateral sclerosis; AUC, area under the curve; CSF, cerebrospinal fluid; CI, confidence interval; GFAP, glial fibrillary acidic protein; NfL, neurofilament light chain; p-tau181, plasma phosphorylated tau protein 181.

**Table 5 ijms-24-13976-t005:** Clinical and neuropsychological features in AD/ALS and not-AD/ALS patients.

	AD-ALS (n = 9)	Not-AD/ALS (n = 147)	*p* Values
MMSE scores *, n	28.0 (22.5–29.0), 5	29.0 (28.0–30.0)	**0.03 ^†^**
MMSE scores (age- and education-adjusted) *, n	25.70 (22.90–29.36), 5	28.16 (26.70–28.99), 108	0.35 ^†^
ECAS total scores (age- and education-adjusted) *, n	89.19 (79.91–112.0), 4	108.4 (96.81–116.80), 84	0.16 ^†^
ECAS total scores (equivalent scores) *, n	2.0 (1.25–3.5), 4	4.0 (3.0–4.0), 84	**0.04 ^†^**
ECAS ALS-specific scores (age- and education-adjusted) *, n	64.44 (61.25–84.42), 4	79.87 (72.56–86.08), 84	0.21 ^†^
ECAS ALS-specific scores (equivalent scores) *, n	2.0 (2.0–3.5), 4	4.0 (3.0–4.0), 84	0.057 ^†^
ECAS ALS-nonspecific scores (age- and education-adjusted) *, n	24.77 (18.42–27.82), 4	27.34 (24.10–30.65), 84	0.20 ^†^
ECAS ALS-nonspecific scores (equivalent scores) *, n	3.0 (1.25–4.0), 4	4.0 (2.0–4.0), 84	0.63 ^†^
Visual short-memory test (age- and education-adjusted) *, n	15.80 (14.18–17.97), 5	19.70 (17.43–20.90), 109	**0.01 ^†^**
Visual short-memory test (equivalent scores) *, n	1.0 (1.0–2.5), 5	3.0 (2.0–4.0), 109	**0.04 ^†^**
ALS-CN ^§^ (%)	4/6 (66.7)	73/122 (59.8)	>0.99 ^¶^
ALSci ^§^ (%)	0/6 (0)	7/122 (5.7)	>0.99 ^¶^
ALSbi ^§^ (%)	0/6 (0)	21/122 (17.2)	0.58 ^¶^
ALScbi ^§^ (%)	1/6 (16.7)	8/122 (6.5)	0.36 ^¶^
ALS-FTD ^§^ (%)	1/6 (16.7)	13/122 (10.6)	0.50 ^¶^

*: Expressed as median (interquartile range); ^†^: Mann–Whitney test; ^§^: expressed as a fraction of total patients with available data; ^¶^: Fisher’s test. Significant *p*-values are reported in bold. Abbreviations: AD, Alzheimer’s Disease; ALS, amyotrophic lateral sclerosis; ALSbi, amyotrophic lateral sclerosis with behavioral impairment; ALScbi, amyotrophic lateral sclerosis with combined cognitive and behavioral impairment; ALSci, amyotrophic lateral sclerosis with cognitive impairment; ALS-CN, cognitively normal amyotrophic lateral sclerosis; ECAS, Edimburgh Cognitive and Behavioral ALS Screen; FTD, frontotemporal dementia; MMSE, Mini-Mental State Examination.

**Table 6 ijms-24-13976-t006:** Multivariable Cox regression analysis for survival in ALS patients, including plasma biomarkers.

Variable	HR (95% CI)	*p*-Value
Plasma GFAP	1.73 (1.05–2.87)	**0.032**
Plasma p-tau181	1.09 (1.00–1.19)	**0.042**
Plasma NfL	1.01 (1.00–1.01)	**<0.001**

Data are expressed as Hazard Ratios and 95% CI. Significant *p*-values are reported in bold. Abbreviations: CI, confidence interval; GFAP, glial fibrillary acidic protein; HR, hazard ratio; NfL, neurofilament light chain; p-tau181, plasma phosphorylated tau protein 181.

## Data Availability

The data presented in this study are available on reasonable request from the corresponding author.

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
