# Peer review of "Amyloid-Beta Co-Pathology Is a Major Determinant of the Elevated Plasma GFAP Values in Amyotrophic Lateral Sclerosis"

_ijms, 2023, doi:10.3390/ijms241813976_

Round 1

Reviewer 1 Report

In this manuscript, Parchi and colleagues provide compelling evidence supporting the potential of plasma GFAP as a valuable tool for understanding AD-related pathology in ALS and its impact on cognitive impairment in these patients. The study builds upon previous research that indicated elevated plasma GFAP levels in ALS patients compared to controls. To explore these associations further, the authors conducted an extensive investigation involving 156 ALS patients and 48 controls, examining the relationships between plasma GFAP and clinical variables, as well as other biofluid biomarkers.

A noteworthy finding is the significant differences in cognitive measures between ALS-AD patients and other ALS cases. This highlights the relevance of AD co-pathology in influencing plasma GFAP values in ALS patients.

The data presented in the manuscript strongly support the authors' conclusions. Additionally, the manuscript exhibits a high quality of English language, and the supplementary file is well-prepared. Therefore, I believe this manuscript is suitable for publication in the International Journal of Molecular Sciences.

One minor correction that needs attention pertains to the references. The format is inconsistent, and some journal abbreviations require revision for accuracy and uniformity.

Author Response

Reviewer 1

In this manuscript, Parchi and colleagues provide compelling evidence supporting the potential of plasma GFAP as a valuable tool for understanding AD-related pathology in ALS and its impact on cognitive impairment in these patients. The study builds upon previous research that indicated elevated plasma GFAP levels in ALS patients compared to controls. To explore these associations further, the authors conducted an extensive investigation involving 156 ALS patients and 48 controls, examining the relationships between plasma GFAP and clinical variables, as well as other biofluid biomarkers.

A noteworthy finding is the significant differences in cognitive measures between ALS-AD patients and other ALS cases. This highlights the relevance of AD co-pathology in influencing plasma GFAP values in ALS patients.

The data presented in the manuscript strongly support the authors' conclusions. Additionally, the manuscript exhibits a high quality of English language, and the supplementary file is well-prepared. Therefore, I believe this manuscript is suitable for publication in the International Journal of Molecular Sciences.

We thank the reviewer for his/her positive comments on the manuscript.

One minor correction that needs attention pertains to the references. The format is inconsistent, and some journal abbreviations require revision for accuracy and uniformity.

We thank the reviewer for raising this point.

We have extensively reviewed the references section accordingly.

Reviewer 2 Report

In this paper, the authors examined the levels of glial acidic fibrillary protein (GFAP) in the plasma of amyotrophic lateral sclerosis (ALS) patients in comparison to controls. They conducted a comprehensive analysis involving 156 ALS patients and 48 controls, exploring the connections between plasma GFAP and various clinical variables and other biofluid biomarkers. The findings revealed that plasma GFAP levels were significantly elevated in ALS patients compared to controls, and these levels were linked to measures of cognitive decline. A subgroup of ALS patients (12.8%) demonstrated a positive amyloid status, with 9 of them also displaying tau pathology (A+T+), indicating the presence of ALS-AD co-pathology. ALS-AD patients exhibited higher plasma GFAP levels than A- ALS participants and controls, while the comparison between A- ALS and controls did not achieve statistical significance. Plasma GFAP emerged as a more precise biomarker for identifying ALS-AD co-pathology when compared to plasma p-tau181 and plasma neurofilament light chain. Furthermore, cognitive measures differed between ALS-AD patients and other ALS patients, signifying the influence of AD co-pathology on the cognitive characteristics of the disease. Overall, plasma GFAP holds promise as a reliable and accurate biomarker for detecting Alzheimer's disease co-pathology in ALS patients, thereby enhancing our understanding of the cognitive aspects of the disease.

Comments:

1.     The authors should include a section detailing the specific methods and techniques used for measuring plasma GFAP levels. This information will provide readers with a better understanding of the technical aspects of the study.

2.     It is essential for the authors to clarify the statistical significance of the slightly increased plasma GFAP levels observed in females compared to males among ALS patients. Additionally, discussing the potential gender-specific implications of this finding would add valuable insights to the study.

3.     The authors should discuss potential confounding factors, including medication use and comorbidities, that may influence plasma GFAP levels in ALS patients. It is important to address how these factors were controlled for in the analysis to ensure the validity of the study's conclusions.

4.     The authors should acknowledge the limitations arising from the lack of neuropathological correlates of GFAP elevation in the cohort. Additionally, they may consider suggesting future research directions, such as conducting neuropathological studies, to validate the observed associations.

5.     The rationale for selecting specific plasma biomarkers (NfL and p-tau181) for covariate analysis in survival analysis should be clarified by the authors. Furthermore, they should discuss how these biomarkers interact with plasma GFAP in predicting disease progression, providing more insight into their combined predictive value.

Author Response

Reviewer 2

In this paper, the authors examined the levels of glial acidic fibrillary protein (GFAP) in the plasma of amyotrophic lateral sclerosis (ALS) patients in comparison to controls. They conducted a comprehensive analysis involving 156 ALS patients and 48 controls, exploring the connections between plasma GFAP and various clinical variables and other biofluid biomarkers. The findings revealed that plasma GFAP levels were significantly elevated in ALS patients compared to controls, and these levels were linked to measures of cognitive decline. A subgroup of ALS patients (12.8%) demonstrated a positive amyloid status, with 9 of them also displaying tau pathology (A+T+), indicating the presence of ALS-AD co-pathology. ALS-AD patients exhibited higher plasma GFAP levels than A- ALS participants and controls, while the comparison between A- ALS and controls did not achieve statistical significance. Plasma GFAP emerged as a more precise biomarker for identifying ALS-AD co-pathology when compared to plasma p-tau181 and plasma neurofilament light chain. Furthermore, cognitive measures differed between ALS-AD patients and other ALS patients, signifying the influence of AD co-pathology on the cognitive characteristics of the disease. Overall, plasma GFAP holds promise as a reliable and accurate biomarker for detecting Alzheimer's disease co-pathology in ALS patients, thereby enhancing our understanding of the cognitive aspects of the disease.

We thank the reviewer for his/her positive comments.

Comments:

  1. The authors should include a section detailing the specific methods and techniques used for measuring plasma GFAP levels. This information will provide readers with a better understanding of the technical aspects of the study.

We used the single molecule array (SIMOA), an established, validated digital ELISA technique. We believe that further details on this technique are unnecessary for such a paper. Nevertheless, we added the specific of the used kit and a reference for readers unfamiliar with this methodology.  

  1. It is essential for the authors to clarify the statistical significance of the slightly increased plasma GFAP levels observed in females compared to males among ALS patients. Additionally, discussing the potential gender-specific implications of this finding would add valuable insights to the study.

We thank the reviewer for this comment. We agree with him/her that the impact of gender on the distribution of plasma GFAP values deserves further attention.

We have implemented the discussion (lines 308-317) with some comments on the topic.

  1. The authors should discuss potential confounding factors, including medication use and comorbidities, that may influence plasma GFAP levels in ALS patients. It is important to address how these factors were controlled for in the analysis to ensure the validity of the study's conclusions.

We thank the reviewer for raising this point.

We recognize that the impact of comorbidities and medication use on the biomarker levels should be further examined. Unfortunately, we do not have this data available for most of our ALS patients. Therefore, we added a sentence to the study limitations section to raise this point (lines 338-340).

However, we also stated that none of our ALS patients was under Riluzole at the time of sampling. We added a sentence in the Methods section (line 403).

  1. The authors should acknowledge the limitations arising from the lack of neuropathological correlates of GFAP elevation in the cohort. Additionally, they may consider suggesting future research directions, such as conducting neuropathological studies, to validate the observed associations.

We agree with the reviewer, and indeed, we already acknowledged the lack of neuropathological examination as the study's main limitation (lines 340-342).

We added a few sentences to highlight the future perspectives on this topic (lines 342-344), as suggested by the reviewer.

  1. The rationale for selecting specific plasma biomarkers (NfL and p-tau181) for covariate analysis in survival analysis should be clarified by the authors. Furthermore, they should discuss how these biomarkers interact with plasma GFAP in predicting disease progression, providing more insight into their combined predictive value.

We thank the reviewer for raising this point. We decided to run a separate multivariable survival analysis including other plasma biomarkers (which are well-established prognostic factors in ALS) to establish whether plasma GFAP provides a specific contribution to predicting survival in ALS patients. We believe that prognostic estimates based on different plasma biomarkers, each reflecting a different pathogenetic process, may have an added value in ALS patients.

We have implemented the manuscript’s text following the reviewer’s suggestions (lines 262-268 and 450-452).